# Disruption of the mRNA m⁶A writer complex triggers autoimmunity in Arabidopsis

**Carey L. Metheringham** [ORCID]ᵒ, **Anjil K. Srivastava** [ORCID]ᵒ, **Peter Thorpe** [ORCID]ᵒ, **Ankita Maji**, **Matthew T. Parker**ᵁ, **Geoffrey J. Barton** [ORCID], **Gordon G. Simpson** [ORCID]*

School of Life Sciences, University of Dundee, Dundee, United Kingdom

ᵒ Contributed equally
ᵁ Current Address: Max Planck Institute for Plant Breeding Research, Cologne, Germany.
* g.g.simpson@dundee.ac.uk

## Abstract

Distinguishing self from non-self is crucial to direct immune responses against pathogens. Unmodified RNAs stimulate human innate immunity, but RNA modifications suppress this response. mRNA m⁶A modification is essential for *Arabidopsis thaliana* viability. However, the molecular basis of the impact of mRNA m⁶A depletion is poorly understood. Here, we show that disruption of the Arabidopsis mRNA m⁶A writer complex triggers autoimmunity. Most gene expression changes in m⁶A writer complex *vir-1* mutants grown at 17°C are explained by defence gene activation and are suppressed at 27°C, consistent with the frequent temperature sensitivity of Arabidopsis immunity. Accordingly, we found enhanced pathogen resistance and increased premature cell death in *vir-1* mutants at 17°C but not 27°C. Global temperature-sensitive mRNA poly(A) tail length changes accompany these phenotypes. Our results demonstrate that autoimmunity is a major phenotype of mRNA m⁶A writer complex mutants, with important implications for interpreting the role of this modification. Furthermore, we open the broader question of whether unmodified RNA triggers immune signalling in plants.

## Author summary

Genes are transcribed into RNA, and some RNAs are chemically modified in ways that ultimately influence gene function. The most frequently occurring modification of messenger RNA is methylation of adenosine at the N6 position (denoted as m⁶A). The role of m⁶A is context and species-specific. Mutation of components of the mRNA m⁶A writer complex in the model plant *Arabidopsis thaliana* results in embryo lethality. However, what makes mRNA m⁶A modification essential in Arabidopsis is currently unknown. In this study, we asked what changes in gene expression occurred in viable Arabidopsis mutants that had significantly reduced mRNA m⁶A levels. We found that the most prominent

**Data availability statement:** Code availability Source code, R notebooks and Snakemake (Mölder et al, 2021 at 10.12688/f1000research.29032.3) pipelines are available at github.com/bartongroup/m6a_arabidopsis_autoimmunity. Data Availability Illumina FASTQ and ONT FAST5 files for the 17ºC and 27ºC datasets are deposited in the ENA under accession code PRJEB85795. ONT FAST5 for the *te234* and *cpsf30-yth* datasets are deposited in the ENA under accession codes PRJEB85859 and PRJEB85860 respectively. Source data for LC-MS, flood inoculation and image staining are available in supporting files.

**Funding:** This work was supported by funding from UKRI | Biotechnology and Biological Sciences Research Council (BBSRC): (awards BB/V010662/1 and BB/M010060/1 to G.G.S. and G.J.B.; and award BB/W007673/1 to G.G.S.). The funders had no role in study design, data collection and analysis, decision to publish, or preparation of the manuscript.

**Competing interests:** The authors have declared that no competing interests exists.

changes in gene expression fell into the categories of defence or immune response. Defence gene expression patterns are frequently temperature sensitive in Arabidopsis. Remarkably, we found that 91% of the genes upregulated in mRNA m6A mutants at 17 °C were not upregulated at 27 °C. Therefore, the main finding of this study is that mRNA m6A mutants exhibit autoimmunity. This raises the question of how defence signalling is activated in mRNA m6A mutants. Furthermore, to understand the direct role of mRNA m6A, approaches that consider the widespread indirect changes in autoimmune gene expression will be required.

## Introduction

Distinguishing self from non-self is crucial to ensuring organisms specifically target immune responses against pathogen infection. RNA modifications provide one layer by which this distinction is made in humans [1,2]. Katalin Karikó, Drew Weissman and colleagues revealed that unmodified RNA stimulates the mammalian innate immunity system by activating the Toll-like receptors (TLRs) TL3, TL7 and TL8, but incorporating modified nucleosides into RNA ablated this activity [2]. Consistent with this, modified nucleosides have been critical in mRNA therapeutics development, and the first mRNA-based vaccines for COVID-19 were based on 1-methyl pseudouridine-containing mRNA [3]. A broader set of factors beyond TLRs function in RNA sensing in humans using RNA structure (including modifications), localisation and availability to distinguish self from non-self [4]. Precision in this process is important because chronic activation of nucleic acid sensing pathways in humans is associated with autoimmune and autoinflammatory conditions [5].

The most abundant internal modification of mRNA is the methylation of adenosine at the N6 position (m6A) [6]. Null mutations that eliminate the activity of the corresponding N6 methyladenosine methyltransferase METTL3 are embryonically lethal in mouse, demonstrating that this modification can play essential roles in biology [7]. A complex of proteins functions with METTL3 to modify mRNA. Orthologs of the human writer complex components, METTL3, METTL14, VIRILIZER, ZC3H13, WTAP and HAKAI, are conserved in Arabidopsis and required for mRNA m6A modification [8,9,10]. Arabidopsis null mutations in each of these components (except the HAKAI and ZC3H13 orthologs) are not viable, and where they exist, hypomorphic alleles have pleiotropic developmental defects [8,10,11,9]. In humans and Arabidopsis, m6A is predominantly written into the terminal exon of mRNA in a preferred context characterised by the DRm6ACH consensus [12,6]. In Arabidopsis, mRNA m6A is also found in a (G)GAU sequence context, albeit less frequently [12,13,14,15].

Reader proteins recognise RNA m6A modifications and ultimately influence mRNA processing and fate [16,17]. The best-characterised m6A reader proteins have a YTH domain, which binds m6A through a cage of aromatic amino acids. Plants and apicomplexans are unique with respect to m6A readers because the conserved CPSF30

component of the cleavage and polyadenylation complex, which binds the AAUAAA poly(A) signal [18,19], has a YTH domain [20]. Consistent with this, a major impact of m6A loss on pre-mRNA processing in Arabidopsis writer complex mutants is disrupted poly(A) site usage [12,21,11]. Not all m6A effects are mediated by YTH reader domains. m6A can affect RNA structure [22,23] and, through an m6A switch mechanism, influence the association of specific RNA-binding proteins with transcripts in an m6A-dependent manner [24,25,26].

A multilayered innate immunity system mediates defence against pathogens in flowering plants [27]. The first layer consists of trans-membrane receptor proteins called pattern-recognition receptors (PRRs) that detect pathogens in the external environment and signal an immune response known as pattern-triggered immunity (PTI). The second layer comprises networks of proteins that detect pathogen effectors and their activity inside plant cells and is known as effector-triggered immunity (ETI). ETI is mainly mediated by nucleotide binding/leucine-rich repeat (NLR) receptors. Cross-talk between PTI and ETI potentiates the immune response [28,29,30,31]. Diverse immune receptors (but not all) converge on shared signalling complexes, such as those containing EDS1, to promote an immune response [27]. In cells infected by pathogens, this can trigger programmed cell death, called the hypersensitive response. Immune responses in neighbouring cells are also activated, but the gene expression pattern differs from those in infected cells [32,33]. A concentration gradient of the hormone salicylic acid (SA) between the sites of infection and neighbouring cells controls the hypersensitive response and the massive expression of defence genes such as *PATHOGEN RESPONSIVE 1* (*PR1*) in surrounding cells and systemic acquired resistance in distal tissues [34,35,36].

PRRs and NLRs are encoded by some of the largest and most rapidly evolving gene families in plants [37]. This diversity corresponds to selective pressure not only for pathogen defence but also to dampen immune responses in the absence of infection, reflecting trade-offs between the benefits of disease resistance and the costs of sustained immune responses on development. Like humans, plants can develop autoimmune conditions [38,39,40,41]. In Arabidopsis, these manifest as compromised development and premature cell death, visible as leaf lesions. Some of the clearest examples of autoimmunity emerge in crosses between different Arabidopsis accessions [42,41]. This phenomenon, observed in the first or later generations of plant hybrids, is known as hybrid necrosis due to the severe pleiotropic symptoms that compromise development and viability [43]. A recurring explanation for hybrid necrosis is simple non-compatible interactions between specific NLRs or other defence genes that activate immune response pathways [41]. Naturally occurring genetic variation [44] or induced mutations [39,40] also trigger Arabidopsis autoimmunity. For example, gain-of-function mutations that either stabilise the expression or autoactivate NLRs can cause autoimmunity [45,46], and so can the disruption of signalling pathways that are either involved in or sensed by defence responses [39]. Arabidopsis autoimmunity is frequently suppressed by either elevated temperature or relatively high humidity, a property that has facilitated the study of autoimmune genotypes [38].

Although the Arabidopsis mRNA m6A writer complex is essential for viability, the gene expression changes that explain this are unknown. Here, we asked what groups of genes are affected when the mRNA m6A writer complex is disrupted. We discovered that immune response genes comprise the major class of altered gene expression, but consistent with the frequently temperature-sensitive nature of Arabidopsis immunity, this response was suppressed when plants were grown at elevated temperatures. Furthermore, Arabidopsis mRNA m6A writer complex mutants display temperature-sensitive increased resistance to pathogen infection and increased levels of premature cell death. Therefore, autoimmunity is a major phenotype of Arabidopsis mRNA m6A writer complex mutants. In contrast to cases of hybrid necrosis and some other autoimmune mutants, visible developmental defects of mRNA m6A writer complex mutants were not rescued by growth at elevated temperatures, revealing that the impact of defective mRNA m6A modification on autoimmunity and development is separable. Therefore, as with humans, RNA modifications in Arabidopsis may contribute to distinguishing self from non-self. Our findings suggest that uncovering how the disruption of the mRNA m6A writer complex triggers defence gene expression is fundamental to understanding the role of this RNA modification in plant biology.

## Results

### Immune gene expression is activated in mRNA m6A writer complex mutants

To understand the roles of mRNA m6A in Arabidopsis, we asked what groups of genes were most affected by the loss of function of the m6A writer complex protein, VIRILIZER. We previously characterised gene expression changes in *vir-1* mutants using a combination of Illumina RNA sequencing (RNA-seq) and Oxford Nanopore Technologies direct RNA sequencing (ONT DRS) [12]. We analysed three genotypes with different VIR activity: a wild-type Col-0 control, a hypomorphic Arabidopsis *vir-1* mutant defective in *VIR* function, and a complementation line expressing VIR fused to Green Fluorescent Protein (GFP) (VIR complemented; VIRc) that partly restores VIR activity in the *vir-1* mutant background [8,12]. For each genotype grown in sterile conditions at 22°C, we sequenced RNA purified from seedlings of at least six biological replicates with Illumina RNA-seq and four with ONT DRS (S1 File).

Using the Illumina RNA-seq data, we identified differentially expressed genes between *vir-1* and WT Col-0 by fitting a quasi-likelihood model in edgeR [47] (threshold: adj.p < 0.001, log2FC > 2.0). We found 806 genes significantly upregulated in *vir-1* compared to Col-0 and 349 genes significantly downregulated (S2 File). We examined GO (gene ontology) term distribution among the differentially expressed genes using gProfiler [48]. The most significantly enriched GO terms were related to response to external stimuli and defence. For example, 92 of the 597 upregulated genes with GO term annotation were annotated with the biological process 'defense response' (GO:0006952), a significant enrichment (adjusted p = 1.32x10$^{-8}$) compared to the background of all genes (Fig 1A and S2 File).

To examine the global expression trends for defence-related genes, in a different way, we identified 1033 genes that included 'defence' or 'defense' in their TAIR annotation description [49]. Of these genes, 86 were differentially expressed in at least one condition. Plotting the zero-centred fold change of these genes shows the extent of expression recovery in the VIRc complementation line (Fig 1B). For example, expression of the defence marker gene *PR1* (AT2G14610) was 311-fold (8.28 log2FC) higher in *vir-1* than Col-0 but restored to similar levels as Col-0 in the VIRc complementation line (Fig 1C). The upregulation of *PR1* in *vir-1* is also detected in the orthogonal ONT DRS data (S1A and S1B Fig). We next asked if other m6A writer mutants had elevated *PR1* expression. We analysed ONT DRS data of *fip37–4* mutants that disrupt the Arabidopsis m6A writer complex ortholog of WTAP [21]. Genes which are significantly upregulated in *fip37–4* mutants are significantly enriched for GO terms related to defence (S3 File). Like *vir-1* mutants, *fip37–4* mutants have elevated *PR1* expression (Fig 1F). In contrast, *PR1* is not upregulated in *fiona1* mutants that disrupt the Arabidopsis N6 methyladenosine methyltransferase METTL16 ortholog that adds m6A to U6 snRNA [21] (Fig 1D). We conclude that a phenotype of Arabidopsis mRNA m6A writer complex mutants is the upregulation of genes involved in defence signalling.

### Loss of the mRNA m6A writer complex triggers temperature-sensitive autoimmunity

The upregulation of defence response genes in *vir-1* raised the question of whether autoimmunity might explain the gene expression changes and pleiotropic developmental phenotypes of mRNA m6A writer complex mutants. Arabidopsis autoimmunity is frequently temperature-sensitive, with autoimmune phenotypes suppressed at elevated ambient temperatures [38]. Therefore, to address whether the gene expression changes we detected in *vir-1* mutants reflected an autoimmune response, we compared the gene expression profiles of *vir-1* and WT Col-0 seedlings grown in sterile conditions at 17°C and 27°C. We used a combination of Illumina RNA-seq and ONT DRS to analyse gene expression. We performed four biological replicates with each RNA sequencing technology, genotype and temperature treatment. The resultant sequencing statistics are detailed in S1 File.

We analysed the Illumina RNA-seq data using a quasi-likelihood model (glmQLFit) in edgeR [47] and identified genes differentially expressed between *vir-1* and WT Col-0 at each temperature. This revealed 1215 genes which were significantly upregulated (adj.p < 0.001, log2FC > 2.0) in *vir-1* at 17°C (S4 File). Remarkably, 91% of these genes (1103) were not significantly upregulated in *vir-1* at 27°C (Fig 2A), revealing that most gene expression changes in *vir-1* are

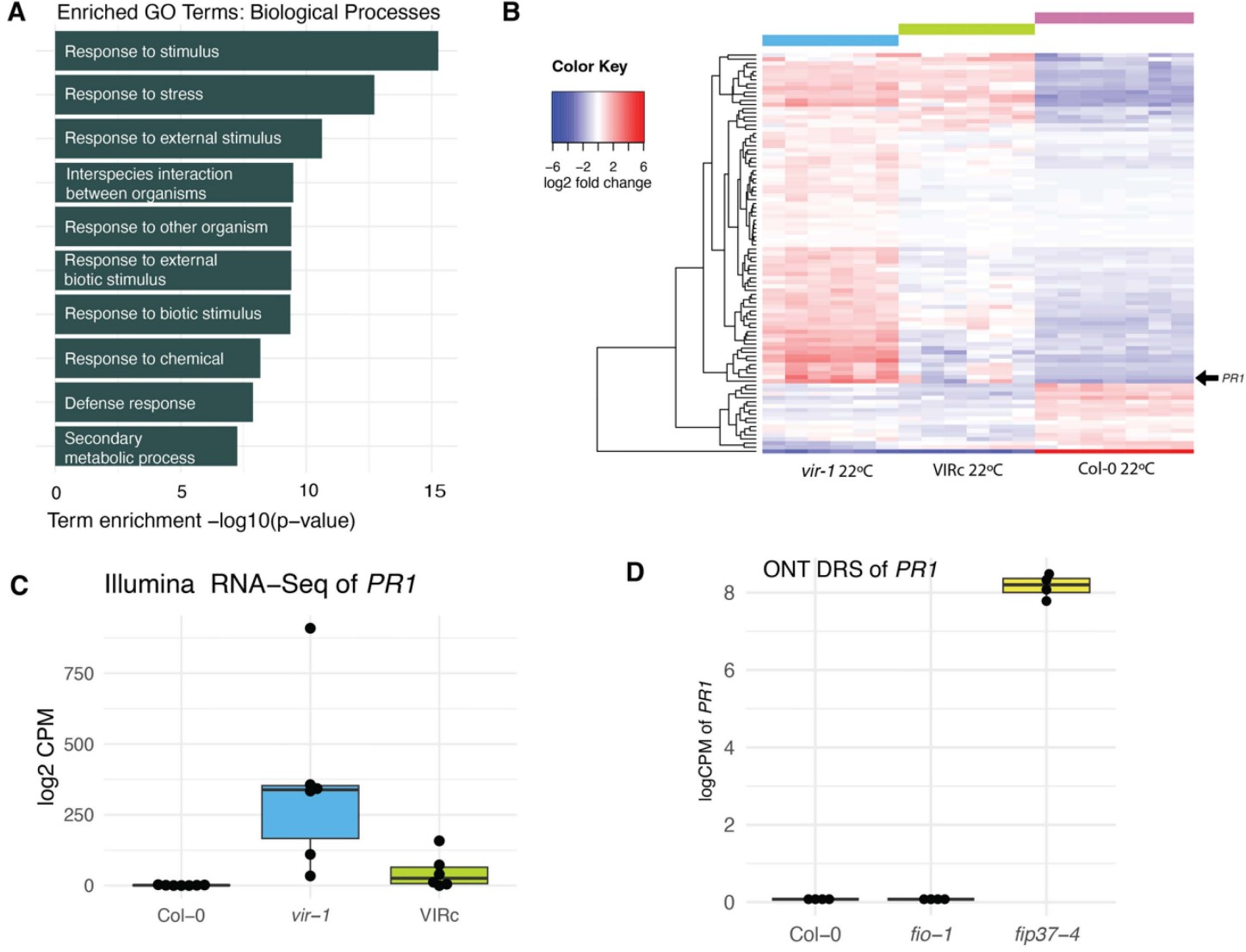

**Fig 1. Immune gene expression is activated in m⁶A writer complex mutants. A)** Top 10 gene ontology terms enriched in the set of 597 genes significantly upregulated (FDR<0.001, log2FC>2.0) in *vir-1* mutants compared to Col-0 WT at 20°C. **B)** Heatmap showing the TMM-FPKM normalised log2 centred fold expression between *vir-1*, Col-0 and VIRc for all genes with TAIR annotations including the term 'defense/defence'. *PR1* (AT2G14610) is highlighted with an arrow. **C)** Normalised log2 counts per million of *PR1* (AT2G14610) in Col-0 (n=7), *vir-1* (n=6) and VIRc (n=6) in Illumina RNA-seq. Boxes represent the interquartile range of the logged values. **D)** Normalised log2 counts per million of *PR1* (AT2G14610) in Col-0, *fip37-4* and *fio1-1* at 20°C in ONT DRS, showing the upregulation of *PR1* in *fip37-4*.

temperature-sensitive. Principal component analysis (PCA) separates the biological replicates by genotype and temperature. The first component, which explains 40% of the variance, captures gene expression changes specific to *vir-1* at 17°C. In contrast, *vir-1* and Col-0 are indistinguishable at 27°C in this component (Fig 2B). Likewise, correlation matrix analysis reveals that the gene expression features of *vir-1* mutants grown at 17°C are the most distinct among all the datasets (Fig 2C). We found elevated expression of *PR1* in *vir-1* grown at 17°C, just as we had previously seen at 22°C, but *PR1* expression was at similar levels to WT Col-0 in *vir-1* grown at 27°C (Fig 2D). We also detected this differential *PR1* expression pattern with orthogonal ONT DRS data (S2A Fig) and RT-qPCR (S2B Fig).

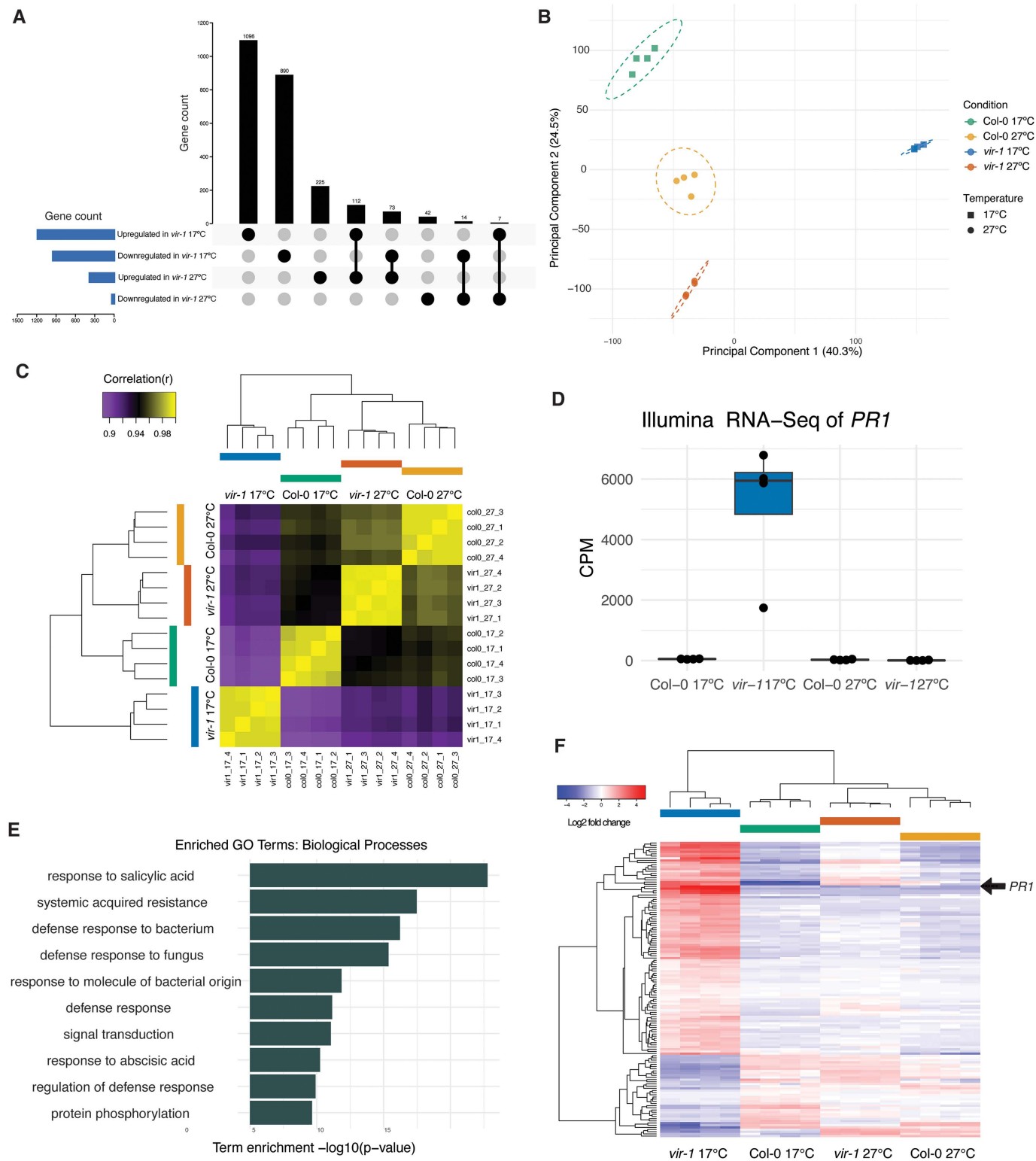

**Fig 2. Loss of m⁶A triggers a temperature-sensitive autoimmune response. A)** Upset plot showing the overlap in significantly upregulated and downregulated genes (log2FC +/- 2.0 FDR < 0.001) in *vir-1* at 17°C compared to Col-0 at 17°C and *vir-1* at 27°C compared to Col-0 at 27°C. **B)** Principal component analysis showing the clustering of samples by experimental condition (including genotype) and temperature. **C)** Correlation matrix and

hierarchical clustering of expression profiles for each condition. The clustering shows that gene expression patterns are distinct for all conditions. In addition, biological replicates within conditions cluster together. However, the gene expression patterns detected in *vir-1* separate as the most different of all possible comparisons. **D)** Significant upregulation of *PR1* (AT2G14610) in *vir-1* at 17°C, shown by a boxplot of normalised expression (log 2 counts per million) in Illumina RNA-seq data (n = 4 samples per genotype). **E)** Most enriched GO terms among genes significantly upregulated (FDR < 0.001, log2FC<2.0) in *vir-1* at 17°C compared to the average of: Col-0 17°C, 27°C and *vir-1* at 27°C. Source data available in S6 File. **F)** Heatmap showing the TMM-FPMK normalised log2 centred fold change for all differentially regulated genes in *vir-1* at 17°C (log2FC +/- 2.0 FDR < 0.001) with TAIR annotations including 'defense/defence' for conditions *vir-1* at 17°C, 27°C, Col-0 at 17°C and 27°C. *PR1* (AT2G14610) is highlighted with an arrow.

We used a generalised linear model (GLM) to model all conditions simultaneously and identify differential gene expression specific to *vir-1* at 17°C. The GLM design contrasts *vir-1* at 17°C minus the average of Col-0 at 17°C and 27°C and *vir-1* at 27°C. This model identified 931 genes with significantly increased expression (adj.p < 0.001, log2FC>2.0) in *vir-1* at 17°C compared to the other conditions (S2C Fig and S5 File). GO-term analysis revealed that the biological processes most significantly enriched in genes upregulated in *vir-1* at 17°C were related to defence responses (Fig 2E and S6 File). The defence annotation GO terms were similar in describing the gene expression changes previously detected in *vir-1* grown at 22°C (S2D Fig). As an orthogonal approach, we analysed protein domain enrichment using DAVID [50] (S6 File). The most significantly enriched protein domains in genes upregulated in *vir-1* at 17°C included RLP23-like, which is found in receptor-like kinases that function as PRR proteins (15-fold enrichment), Defensin_plant, a domain found in highly expressed marker proteins of defence (10-fold enrichment), and WRKY_plant (6-fold enrichment), which is found in WRKY transcription factors that frequently control defence response transcription networks [51]. We used AME (Analysis of Motif Enrichment) from the MEME Suite to identify enriched motifs in promoter regions of genes upregulated in *vir-1* at 17 °C compared to Col-0 at 17 °C. There was significant enrichment (p < 0.01) of 15 transcription factor binding motifs from the DAP-seq database [52], including NAC, MYB-related, and W-Box (WRKY binding) motifs, consistent with the enrichment of WRKY protein domains in the genes upregulated in *vir-1* at 17 °C (S7 File). The genes upregulated in *vir-1* at 17 °C versus all other conditions were enriched for ATAF1, KAN2, AT5G56840, and AT2G20400 binding motifs (S7 File). There was no enrichment of any motif in the promoters of genes upregulated in *vir-1* at 27 °C compared to Col-0 at 27 °C. Together, these motif analyses are consistent with an immune state in *vir-1* at 17 °C, but also indicate that other transcription factor classes may contribute to a more complex stress-related transcriptional program.

Next, we used a different approach to ask how defence gene expression was affected by temperature using the GLM analysis. We plotted the zero-centred log2FC normalised gene expression of 136 genes with defence/defense included in their TAIR annotation in *vir-1* at 17°C compared to other conditions. This analysis reveals that the expression level of most genes with a TAIR defence/defense annotation in *vir-1* at 17ºC is at WT Col-0 levels when *vir-1* mutants are grown at 27°C (Fig 2F).

To determine whether the major defence gene expression differences observed in *vir-1* at 17ºC compared to all other conditions might be explained by overlooked pathogen contamination of our experimental material, we examined our RNA-seq data for non-Arabidopsis sequences. Although the poly(A) mRNA purification step used in Illumina RNA-seq may limit the sensitivity of comprehensive bacterial RNA detection, this control should indicate if plant pathogens (including fungal and oomycete species) were a major concern. We used all *vir-1* Illumina RNA-seq data to produce a *de novo* transcriptome assembly and searched the resulting contigs against the GenBank NR (non-redundant) database using BLASTP [53]. No significant enrichment of plant pathogen sequences was found in *vir-1* 17°C samples compared to Col-0 (S8 File).

In summary, by exploiting the established frequent temperature sensitivity of Arabidopsis immunity, we discovered that the major annotation terms associated with the upregulated genes in *vir-1* mutants at 17°C are related to defence. Therefore, at the gene expression level and strikingly dependent on temperature, we conclude that autoimmunity is a major phenotype of the Arabidopsis mRNA m⁶A writer complex mutant, *vir-1*.

## Genes that function in diverse aspects of immunity are upregulated in *vir-1* mutants

Given the global trend of defence gene activation in *vir-1* mutants, we next analysed individual gene expression changes to understand what type of defence genes were affected. Expression of mRNA encoding the master defence transcription factor SARD1 [54,55] was upregulated (AT1G73805: log2FC 4.0) in *vir-1* at 17°C but not 27°C, consistent with the established temperature sensitivity of *SARD1* transcription [56] (Figs 3A and S3A). The expression of mRNA encoding the FLS2 receptor, which detects the bacterial flagellin flg22 peptide [58] and is one of the best-characterised Arabidopsis PRR proteins, was upregulated in *vir-1* at 17°C (AT5G46330: log2FC 3.0) but not at 27°C (vir_27 vs Col-0_27: log2FC 1.3) (Figs 3B and S3B). We detected the upregulation of 15 annotated NLRs - 13 TIR-NLRs, one CC-NLR (*LOV1*) and one RPW8-NLR (*HR4*) at 17°C but not 27°C (S9 File). For example, the TIR-only *TX0* was upregulated (AT1G57630: log2FC 4.1) (Figs 3C and S3C). TX0 can hydrolyse nucleic acids, particularly RNA, and synthesise 2',3',-cAMP/cGMP molecules that ultimately signal cell death in the hypersensitive response [59].

We next asked whether genes previously associated with Arabidopsis autoimmunity were misregulated in *vir-1.* The TIR-NLR *RPS6* (AT5G46470) is recurrently associated with autoimmunity. For example, the extreme phenotypes of Arabidopsis nonsense-mediated RNA decay (NMD) and mitogen-activated kinase mutants have been attributed to *RPS6* [60,61], although the mechanisms involved are not understood [60,61,62]. *RPS6* expression is not significantly altered in *vir-1* RNA-seq data, but ONT DRS analysis indicates that the TIR-only gene located downstream of the *RPS6* locus is upregulated [62] (S3D Fig). The TIR-NLR *SNC1* has been used as a model to understand autoimmune signalling [63]; *SNC1* was not significantly upregulated in *vir-1* at 17°C (log2FC 1.22), but *SIDEKICK3*, a TIR-NLR required for *SNC1*-mediated autoimmunity [64], is one of the most upregulated genes (log2FC 8.32) in *vir-1* at 17°C (S9 File). Finally, we examined *ACD6,* which encodes a multipass transmembrane protein with intracellular ankyrin repeats, that mediates a trade-off between growth and defence [65]. First identified in lab-based mutant screens [66,67], high-activity *ACD6* alleles are frequently found in natural Arabidopsis accessions [68,69,44]. *ACD6* is upregulated in *vir-1* at 17°C (log2FC 6.27), but the expression level is similar to WT Col-0 in *vir-1* grown at 27°C (Figs 3D and S3E). We asked whether the changes in gene expression between *vir-1* and *acd6* mutants were related by re-analysing a recently published *acd6–1* Illumina RNA-seq dataset [57]. We found a subset of differentially expressed genes overlap between these two mutants; 205 upregulated genes and 21 down-regulated genes in common (Fig 3E). However, 931 genes are uniquely upregulated in *acd6–1* and 701 genes are uniquely upregulated in *vir-1* at these thresholds (adj.p < 0.001, log2FC>2.0), indicating that the misregulation of *ACD6* alone does not simply explain *vir-1* autoimmunity gene expression phenotypes.

A group of flowering time genes paralleled the expression of immune response genes. The floral pathway integrator, *FT,* [70] is upregulated in *vir-1* at 17°C but not 27°C (S9 File). In addition, the expression of a group of genes known to function downstream of *FT* in floral development, including *FUL, SOC1, SEP3, SPL4, SPL5, AGL19* and *AGL24* phenocopied defence gene expression patterns (S9 File and S3F-M Fig).

In summary, diverse genes attributed to the different ETI and PTI layers of Arabidopsis innate immunity were upregulated in *vir-1* mutants at 17°C but not 27°C, together with mRNA encoding the SARD1 master defence transcription factor that controls SA-dependent and SA-independent defence responses.

## *vir-1* mutants exhibit temperature-sensitive pathogen resistance and localised cell death

A characteristic of autoimmunity is that plants show enhanced resistance to pathogens because defence gene expression is already upregulated prior to infection. We, therefore, asked whether the global changes in gene expression detected in *vir-1* resulted in a functional impact on pathogen infection. We examined the susceptibility of WT Col-0, *vir-1* and *fls2* (*flagellin sensing 2*) mutants to the biotrophic pathogenic bacterium *Pseudomonas syringae* pv. tomato (*Pto*) DC3000. The *fls2c* mutant allele [71] lacks the receptor for bacterial flagellin and is susceptible to infection, so is a positive control for infection in these experiments. We flooded seedlings grown at 17°C, 21°C and 27°C with *P. syringae* pv. tomato (*Pst*) DC3000. We found that *fls2c* mutants were more susceptible than Col-0 to infection at all temperatures, consistent with

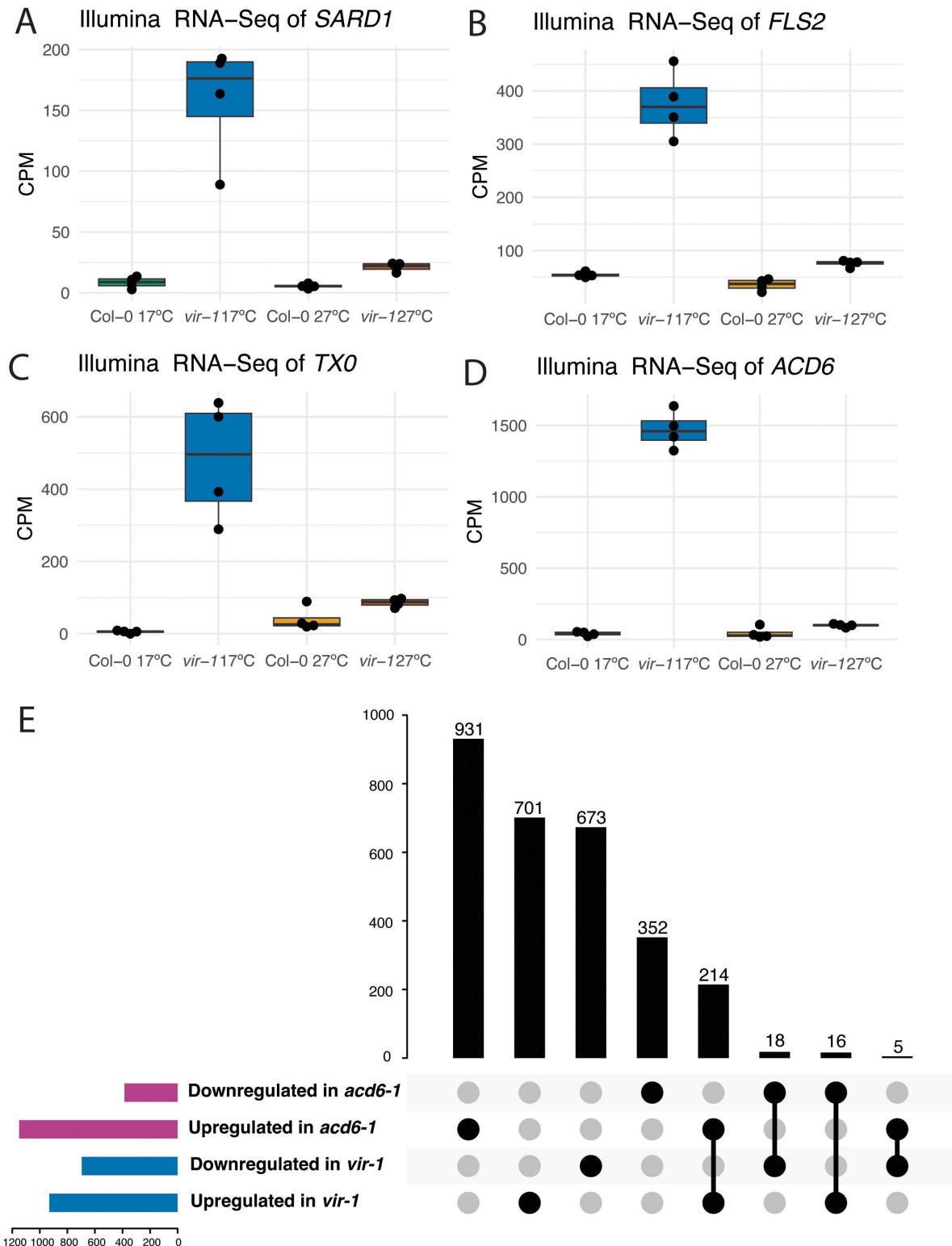

**Fig 3. Defence genes that function in diverse aspects of immunity are upregulated in *vir-1* mutants A) Significant upregulation of *SARD1*** (AT1G73805) in *vir-1* at 17°C, shown by a boxplot of normalised expression (log2 counts per million) in Illumina RNA-seq data (n = 4 samples per condition). **B)** Significant upregulation of *FLS2* (AT5G46330) in *vir-1* at 17°C, shown by a boxplot of normalised expression (log2 counts per million)

in Illumina RNA-seq data (n = 4 samples per condition). **C)** Significant upregulation of *TX0* (AT1G57630) in *vir-1* at 17°C, shown by a boxplot of normalised expression (log2 counts per million) in Illumina RNA-seq data (n = 4 samples per condition). **D)** Significant upregulation of *ACD6* (AT4G14400) in *vir-1* at 17°C, shown by a boxplot of normalised expression (log2 counts per million) in Illumina RNA-seq data (n = 4 samples per condition). **E)** Upset plot showing modest overlap in differentially upregulated genes between *vir-1* at 17°C and previously published *acd6-1* RNA-seq data [57].

previous reports [71] (Figs 4A and S4 and S10 File). In contrast, *vir-1* mutants were more resistant to infection than Col-0 when grown at 17°C and 21°C, but there was no significant difference in infection levels between Col-0 and *vir-1* plants grown at 27°C. Therefore, the temperature-sensitive patterns of defence gene expression detected in *vir-1* convert to a corresponding change in immunity.

Localised premature cell death is a phenotype of plant immune responses to infection, and leaf lesions are a feature of some autoimmune genotypes [39,40]. To investigate whether *vir-1* mutants exhibited elevated levels of cell death

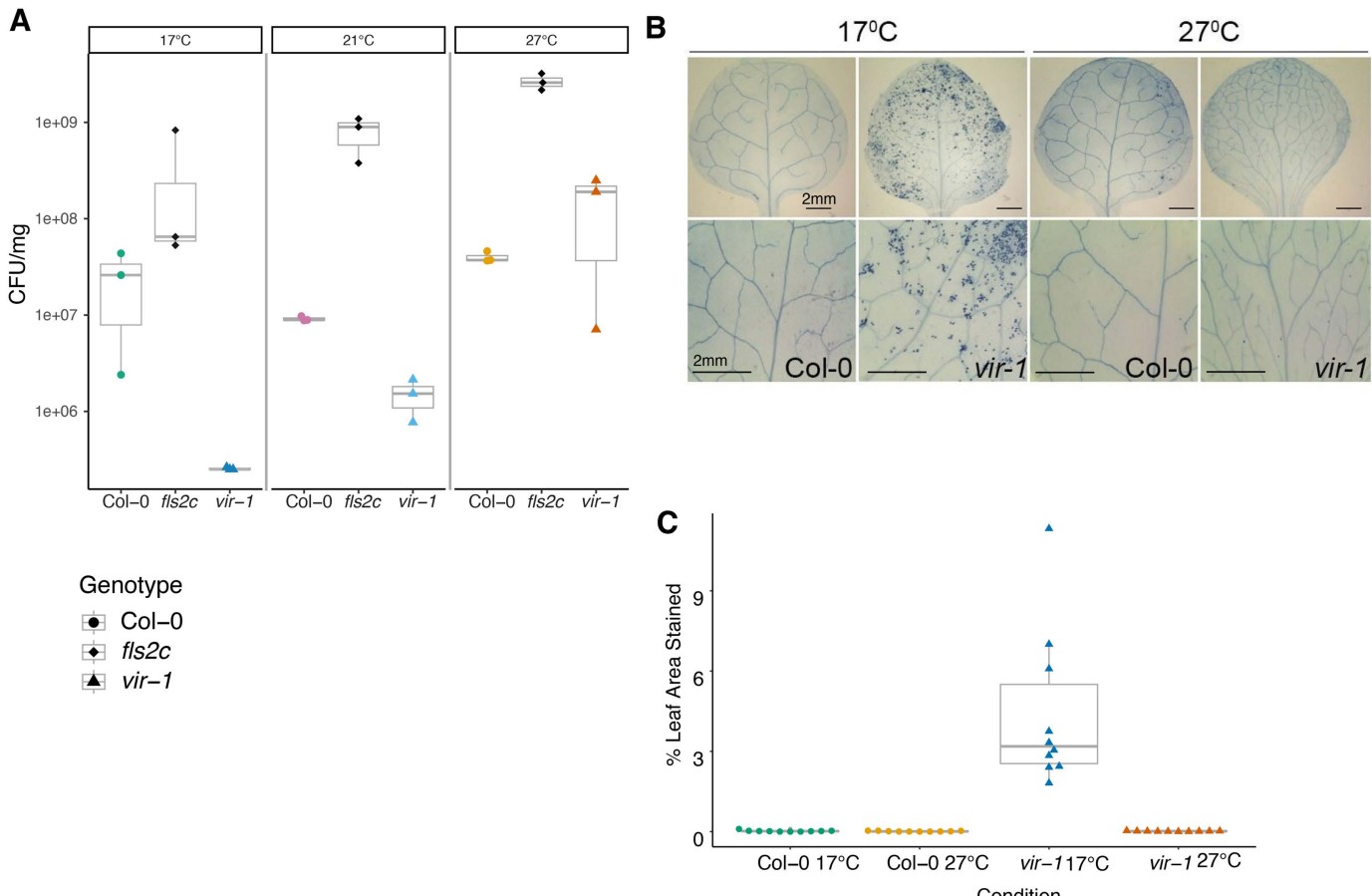

**Fig 4. *vir-1* mutants exhibit temperature-sensitive pathogen resistance and localised cell death.** A) Four-week-old Col-0 WT, *vir-1* and *fls2c* seedlings flood-inoculated with a bacterial suspension of *Pst* DC3000 (5x10$^6$ CFU/ml) and 0.025% v/v Silwet L-77. Bacterial populations were quantified at 3 days post inoculation (dpi) (n = 3 per condition). One way ANOVA tests on each genotype revealed a significant effect of temperature in the *vir-1* genotype (F = 37.23, p = 0.0358) which was not present in Col-0 WT. Source data available in S9 File. This experimental analysis was replicated independently in S4 Fig. B) Trypan blue staining of Col-0 WT and *vir-1* mutant leaves imaged with a Zeiss histology microscope at 10x magnification. C) Estimation of trypan blue staining patterns using ImageJ (n = 10 per condition). Two-way ANOVA revealed significant effects of temperature (F = 22.34, $p = 3.45 \times 10^{-5}$), genotype (F = 22.18, $p = 3.63 \times 10^{-5}$), and their interaction (F = 22.16, $p = 3.66 \times 10^{-5}$). Post hoc comparisons using Tukey's HSD test indicated that *vir-1* at 17°C significantly differed from all other conditions ($p < 0.0001$). Source data available in S10 File.

compared to WT Col-0 in the absence of pathogen infection, we stained seedlings grown in sterile conditions with the vitality marker trypan blue (TB). We recorded microscopy images and quantified the levels of detectable TB staining for 10 individual leaves of each genotype grown in each condition using ImageJ [72]. We detected the highest levels of cell death in *vir-1* grown at 17°C (Fig 4B and 4C and S11 File). However, at 27°C, cell death patterns in *vir-1* were at negligible levels, comparable to those detected in WT Col-0 at 17°C and 27°C. We, therefore, conclude that *vir-1* mutants show elevated levels of premature cell death at 17°C when immune response pathways are autoactivated.

Overall, *vir-1* mutants' temperature-sensitive response to two orthogonal analyses of autoimmunity - enhanced pathogen resistance and increased premature cell death - is consistent with the global patterns of gene expression changes we detect at the RNA level. These findings suggest that the mRNA m⁶A writer complex is required to dampen defence pathway signalling to prevent autoimmunity in the absence of pathogens but not for the defence responses that suppress *P. syringae* pv. tomato (*Pto*) DC3000 infection. We, therefore, conclude that *vir-1* mutants have a temperature-sensitive autoimmune phenotype.

## mRNA m⁶A levels in *vir-1* mutants are not temperature-sensitive

An ethyl methanesulfonate-induced 5' splice site mutation in intron 5 (G + 1 to A) causes the *vir-1* allele, resulting in cryptic splicing events within exon 5 that disrupt the *VIR* open reading frame [8]. Since mutations that disrupt splice sites can be temperature-sensitive [73], we asked if the expression of *VIR* mRNA was restored at 27°C. However, we found no evidence from our RNA-seq data to support this idea (S5A and S5B Fig). We next asked whether mRNA m⁶A levels in *vir-1* mutants were restored to wild-type levels at 27°C. We first used liquid chromatography-tandem mass spectrometry (LC-MS/MS) to analyse the m⁶A/A (adenosine) ratio in poly(A)+ RNA purified from Col-0 and *vir-1* grown at 17°C and 27°C. The level of poly(A)+ RNA m⁶A modification in the hypomorphic *vir-1* allele was reduced to approximately 10% of that detected in Col-0 at both 17°C and 27°C (Fig 5A and S12 File), consistent with previous reports of reduced m⁶A levels in *vir-1* mutants [8,12]. In addition to LC-MS/MS, we used the ONT DRS data to examine mRNA m⁶A levels. We have previously mapped m⁶A in ONT DRS data using the Differr and Yanocomp programs, which depend upon comparing WT

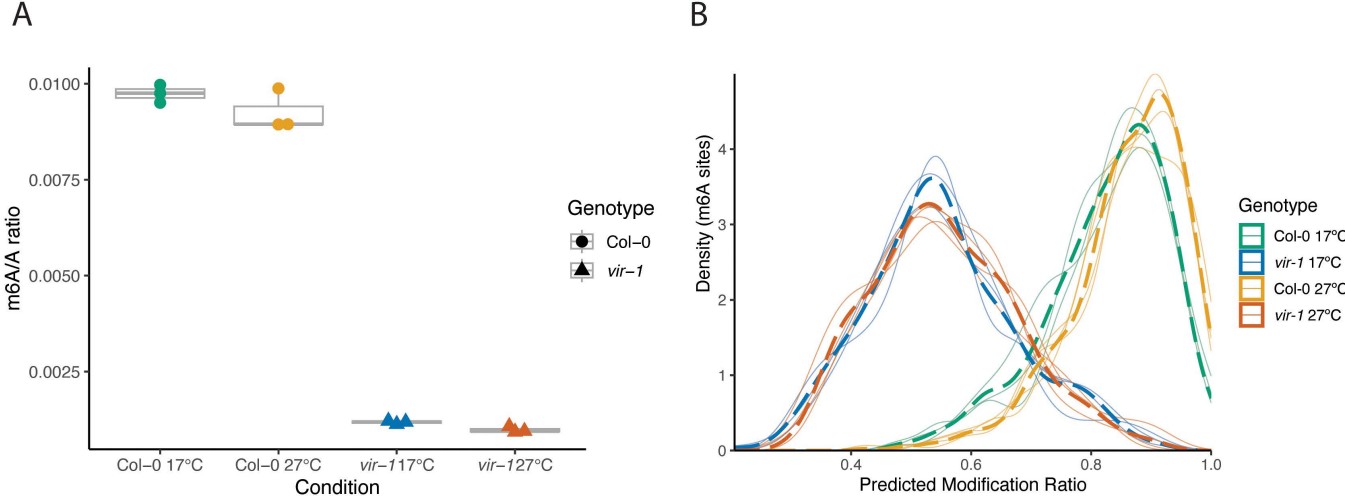

**Fig 5. m⁶A levels in *vir-1* mutants are not temperature-sensitive.** A) LC-MS/MS analysis showing the significant effect of genotype on the m⁶A/A ratio and the lack of significant interaction between genotype and temperature on m⁶A levels (two-way ANOVA; p < 0.001) (n = 3 per condition). Source data available in S11 File. B) Density distribution of the ratio of m⁶A modification per site for all sites with probability modification > 0.9 predicted by m6Anet. Individual replicates are plotted as solid lines, with the combined density of a condition (genotype and temperature) plotted as a dashed line. 3545 sites were predicted to have an m⁶A modified site in at least one Col-0 sample, compared to only 327 in *vir-1*.

and mutant genotypes [12,74]. We supplemented Yanocomp analysis with m6Anet, a neural-network-based method that can call read-level m⁶A stoichiometry without genotype comparison [75]. We found no restoration of m⁶A levels identified by m6Anet (Fig 5B) or Yanocomp analysis (S5C Fig) in *vir-1* at 27°C compared to 17°C. We conclude that the suppression of immune gene expression detected in *vir-1* at 27°C is not explained by an accompanying change in mRNA m⁶A modification.

## The visible developmental defects of *vir-1* mutants are separable from autoimmune gene expression

Arabidopsis autoimmune mutants often show developmental defects that can be rescued by growth at elevated temperatures [38]. We asked whether the impact of autoimmunity might explain the developmental phenotypes of *vir-1*. We examined the development of Col-0 and *vir-1* mutant plants from germination to flowering and seed-set at 17°C and 27°C. We included *acd6–1* as a positive control for an autoimmune mutant compromised in development at 17°C, which appears more like WT Col-0 when grown at 27°C. We found that the short stature and lack of apical dominance characterising the visible developmental phenotypes of *vir-1* mutants were not rescued by growth at 27°C (Fig 6A). However, we were

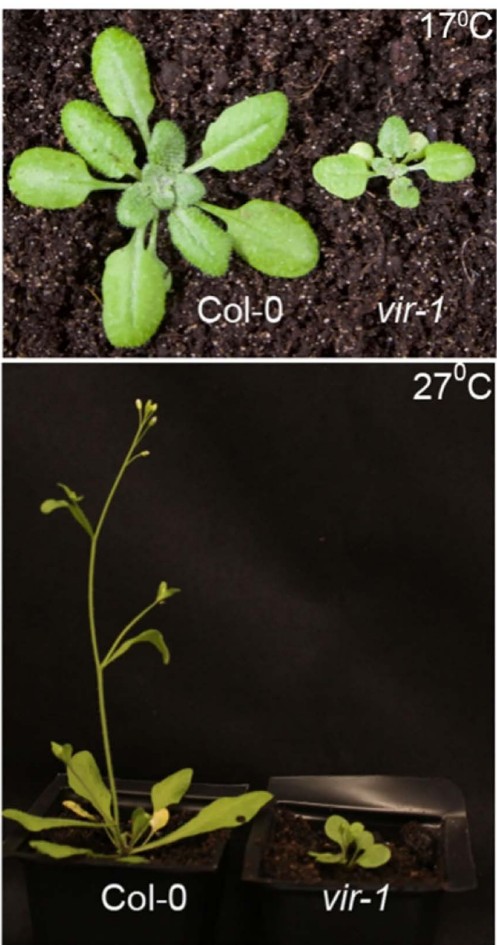

**Fig 6. The primary visible developmental defects of *vir-1* mutants are not rescued by growth at 27°C. A)** Developmental phenotype of Arabidopsis WT Col-0 and mutant *vir-1* grown at 17°C and 27°C. Plants are 28 days old and were grown at the indicated temperatures throughout their development following a 2-day stratification treatment.

able to replicate the previously reported developmental rescue of *acd6–1* mutants at 27°C, compared to 17°C ([S6A Fig]). We, therefore, conclude that the impact of the loss of the mRNA m⁶A writer complex on development and defence gene expression programmes is separable.

Next, we asked if we could detect gene expression changes that might underpin developmental change in *vir-1* mutants. We identified significant gene expression differences consistently affected by the *vir-1* mutation across all our datasets, irrespective of temperature (adj.p < 0.001, log2FC>2.0) ([S13 File]). The flowering time control gene, *FLC*, was the only gene significantly downregulated across all our *vir-1* experimental conditions. Only 58 genes were consistently upregulated. However, the most enriched GO term biological processes for these genes were "defence response", "systemic acquired resistance", and "defence response to other organism" ([S6B Fig] and [S14 File]), indicating that defence gene upregulation remains an important *vir-1* gene expression phenotype. For example, RNA encoding the flavin-dependent monooxygenase, FMO1, was upregulated in *vir-1* relative to WT Col-0 in all tested conditions ([S6C Fig]). FMO1 is a critical regulator of systemic acquired resistance to pathogen infection [76]. We also identified consistent upregulation of the AtNUDT24 nudix hydrolase. AtNUDT24 is uncharacterised, but other members of this protein family function to modify signalling nucleotides in plant defence [59,77] or in RNA decapping and hydrolysis, among other roles [78,79].

Overall, we conclude that although defence gene activation in *vir-1* mutants is suppressed at 27°C, visible developmental defects are not, and autoimmune gene expression programmes remain a component of the *vir-1* mutant phenotype.

## Altered poly(A) tail length distributions are a temperature-sensitive phenotype of *vir-1* mutants

We asked if immune response gene mRNAs were more likely to be m⁶A-modified than other mRNAs by analysing our ONT DRS datasets. It is not possible to determine m⁶A stoichiometry for every immune response gene because most are either not expressed in wild-type Col-0 in the absence of pathogen infection or are expressed at levels below the read count required by m6Anet to predict m⁶A sites. However, for the transcripts with sufficient coverage (greater than 20 mean reads in Col-0 across 17°C and 27°C treatments), we observe 41% fewer m⁶A sites among genes with TAIR annotations including the terms 'defense' or 'defence' compared to the Col-0 transcriptomes as a whole (Fisher's exact test: $p = 6.24 \times 10^{-6}$, odds ratio = 0.59).

To understand how the loss of mRNA m⁶A might trigger an autoimmune response, we next asked if RNA processing was affected in *vir-1* mutants grown at different temperatures. Consistent with our previous reports [12,21], we found that genetic disruption of the mRNA m⁶A writer complex resulted in a global shift to proximal poly(A) site usage in diverse mRNAs ([Fig 7A]). We identified 981 genes with significant differences in poly(A) site usage between Col-0 and *vir-1* at 17 °C and 820 genes with significant differences in poly(A) site usage between Col-0 and *vir-1* at 27 °C. In contrast, only a small number of genes showed significant temperature-sensitive differences in poly(A) site usage, with 48 affected genes in Col-0 and 30 in *vir-1*. Genes with significantly altered poly(A) site usage were not enriched among those genes significantly misexpressed (FDR < 0.001) in *vir-1* at 17°C compared to the other conditions (Fisher's exact test, $p = 0.38$), and there was no correlation between gene expression and the changes in poly(A) site usage observed between Col-0 and *vir-1* at either temperature (Pearson's correlation, $r = 0.014$, $p = 0.23$).

We could identify *vir-1*-dependent poly(A) site shifts at individual genes. For example, at *ISTL2* (AT1G25420), a distal poly(A) site is preferentially used in Col-0 ([Fig 7B]), but two promoter-proximal poly(A) sites are preferentially used in *vir-1*, and the distal site, which was preferentially used in Col-0, is almost unused in *vir-1*. Therefore, consistent with the aggregated data, poly(A) site usage shifts are predominantly evident between Col-0 and *vir-1*, rather than between temperature treatments, at individual genes.

Altered 3' end formation may trigger autoimmunity, but the subsequent signalling might be suppressed at 27°C, so we addressed this question differently. If alternative poly(A) site usage triggered the autoimmune response, it might be mediated by the CPSF30-YTH m⁶A reader. We examined gene expression changes in *cpsf30-yth* mutants using ONT DRS. The *cpsf30-yth* mutants express a truncated protein comprising the N-terminal Zinc-Finger domains that bind the AAUAAA

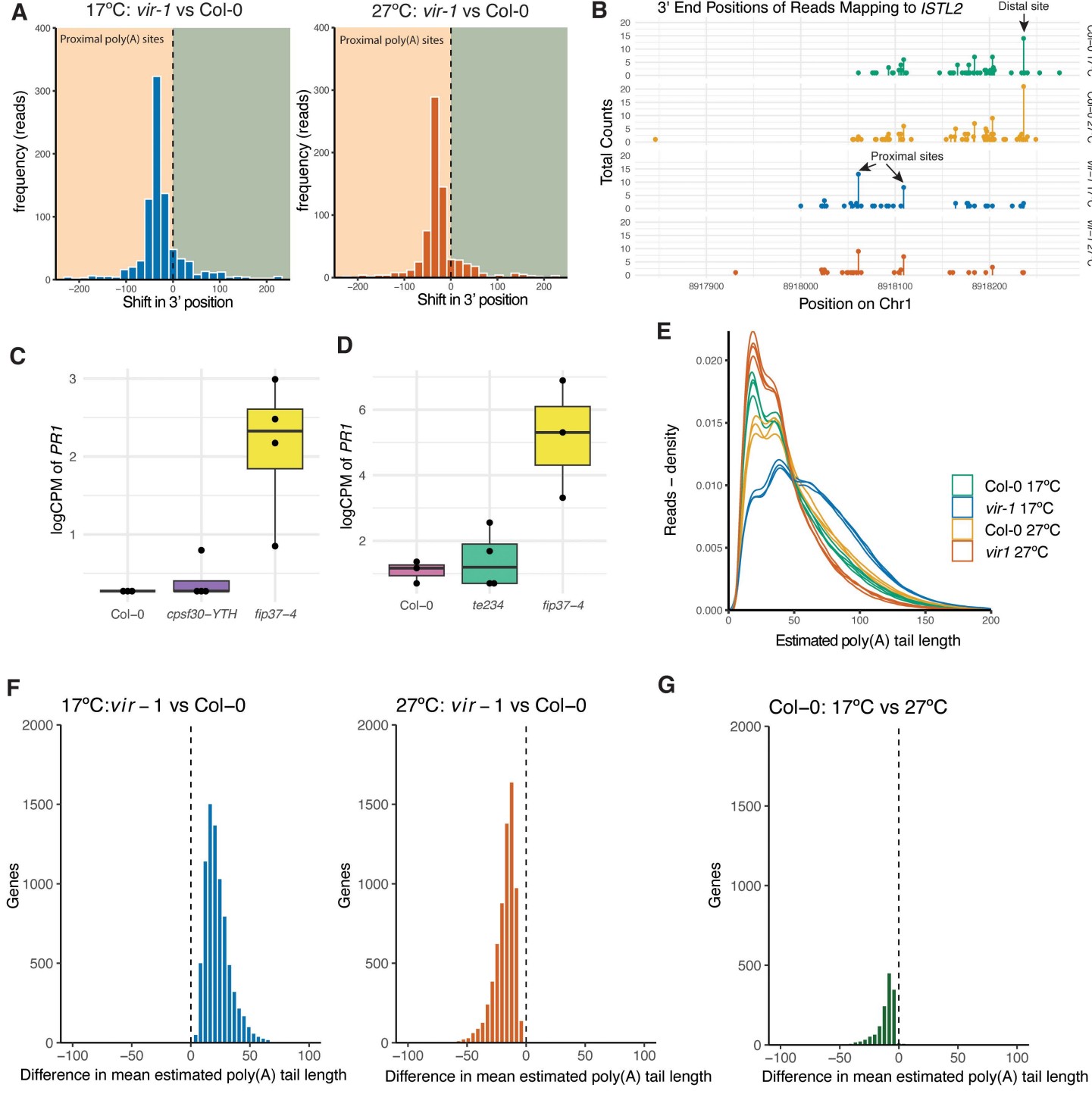

**Fig 7. Poly(A) tail length distributions of *vir-1* are disrupted in a temperature sensitive manner. A)** Shifts towards upstream (promoter-proximal) poly(A) site usage are detected in *vir-1* mutants at 17°C and 27°C compared to Col-0 WT. Source data available in S14 File. **B)** Normalised expression (log 2 counts per million) of *PR1* (AT2G14610) in ONT DRS analysis of *cpsf30-yth* mutants at 20°C (n=4 per condition). **C)** End points of reads mapping to the 3′ end of *ISTL2* (AT1G25420), showing the shift from the predominate use of the poly(A) from a single downstream site in Col-0 samples to two upstream sites in *vir-1* mutants. **D)** Normalised expression (log 2 counts per million) of *PR1* (AT2G14610) in ONT DRS analysis of *te234* triple mutants at 20°C (n=4 per condition). **E)** Density distribution of estimated poly(A) tail lengths in Col-0 and *vir-1* at 17°C and 27°C. The distribution of each replicate is plotted individually. **F)** Histograms depicting the distribution of Wasserstein distance metric for significant changes in mean estimated poly(A) tail

length per gene between Col-0 and *vir-1* at 17°C and 27°C, and between Col-0 at 27°C and 17°C. At 17°C, 13 genes have significantly shorter poly(A) tails in *vir-1*, while 7,894 genes have significantly longer tails. At 27°C, 6,669 genes displayed significantly shorter poly(A) tails in *vir-1*, whereas 7 genes had significantly longer tails. **G)** In Col-0 there are 1,425 genes with significantly shorter mean estimated poly(A) tails at 17°C compared to 27°C. These findings are derived from data pooled across all replicates.

poly(A) signal, but lack the C-terminal YTH domain. We found *PR1* is not upregulated in *cpsf30-yth*, although it is upregulated in *fip37–4* mutants analysed alongside here as a positive control (Fig 7C). Therefore, this combination of data does not provide evidence that altered poly(A) site usage triggers the autoimmunity phenotypes of Arabidopsis mRNA m⁶A writer complex mutants.

The cytoplasmic YTH reader domain proteins likely mediate specific impacts of m⁶A on RNA fate [80]. The most abundant of these are ECT2, 3 and 4 [81]. We analysed a triple mutant in each gene, *te234* [82] using ONT DRS. We found no evidence that *PR1* was upregulated in *te234.* In contrast, *PR1* upregulation was again detected in *fip37–4* mutants included here as a positive control (Fig 7D).

Next, we asked if poly(A) tail length was altered in *vir-1* mutants. We have previously reported a change in poly(A) tail length profiles at specific genes in *vir-1* [12], and we asked if this was a more widespread phenotype. We used the ONT software Dorado to estimate transcript poly(A) tail length in our ONT DRS data. This analysis reveals a characteristic periodicity in the estimated lengths of read poly(A) tails, which likely reflects the footprint of binding of multiple poly(A) binding proteins (PABPs) [83,84]. We found that the distribution of estimated poly(A) tail lengths was markedly different in *vir-1* compared to WT Col-0 - at 17°C; relatively fewer transcripts with short poly(A) tails and more with longer poly(A) tails are detected in *vir-1* compared to WT Col-0 (Fig 7E). At 27°C, *vir-1* mutant mRNA poly(A) tail length profiles are different again, with a distribution more enriched in short poly(A) tails compared to WT Col-0 (Fig 7D). In contrast, the 30 nt poly(A) tail of *Saccharomyces cerevisiae ENOLASE II* RNA, used here as a spike-in calibration standard during ONT DRS library preparation, is consistent across all samples, with an estimated median tail length of 33 nt (S7A Fig). We asked if we could detect changes in estimated poly(A) tail lengths at individual genes. We assessed poly(A) tail length distributions for *GAPC2* (AT1G13440) and *UBQ10* (AT4G05320) transcripts, because they are highly expressed and stably detected across the experimental conditions analysed here. These genes display tail length distribution patterns that closely mirror the global read poly(A) tail length estimates: fewer short poly(A) tails and more long poly(A) tails detected in *vir-1* compared to Col-0 at 17°C and more short poly(A) tails and fewer long poly(A) tails detected in *vir-1* compared to Col-0 at 27°C (S7B and S7C Fig). In a previous study [12], we reported changes in poly(A) tail length distribution of *CAB1* (AT1G29930) transcripts between wild-type Col-0 and *vir-1* mutants. In the current dataset, *CAB1* also exhibits temperature- and genotype-dependent shifts in tail length, with fewer short poly(A) tails and more long poly(A) tails detected in *vir-1* compared to Col-0 at 17 °C (S7D Fig). However, at 27°C, more of the shortest *CAB1* poly(A) tail length distributions are detected in Col-0 than in *vir-1*, suggesting a degree of gene-specific variation.

Changes in the global distributions of poly(A) tail length may be partly due to differences in the sets of genes expressed at detectable levels in the different conditions. To examine the differences in tail length per gene and exclude genes that were only expressed in single conditions, we used the Wasserstein distance metric to quantify shifts in mean per gene tail length distributions between conditions [62]. This analysis identified a shift towards longer mean poly(A) tails in *vir-1* at 17°C compared to Col-0 and a shift towards shorter mean poly(A) tails in *vir-1* at 27°C, consistent with the different distributions of estimated poly(A) tails lengths (Fig 7F and S15 File). Temperature-sensitive changes in mean poly(A) tail length in Col-0 were modest by comparison (Fig 7G).

We next asked if there was any relationship between poly(A) tail length and those mRNAs that are misexpressed in the *vir-1* mutant. This analysis is limited by the fact that many of the genes upregulated in *vir-1* at 17 °C are either expressed at low or undetectable levels in other conditions. For those genes for which we could make this comparison, we found the same pattern of poly(A) tail length distributions for genes expressed at significantly higher levels in *vir-1* at 17 °C

compared to other conditions (S7E Fig). Therefore, global changes in poly(A) tail length are not explained by the immune response genes upregulated in *vir-1* at 17 °C.

Next, we asked whether the shift in mean poly(A) tail length was directly associated with the loss of m⁶A modification. We found that while genes with m6Anet-predicted m⁶A sites had shorter poly(A) tails, the temperature-dependent differences in poly(A) tail length distribution were seen in genes predicted to be either m⁶A-modified or non-modified (S7F Fig).

In summary, our findings reveal global changes in poly(A) tail length distributions as the primary temperature-sensitive mRNA processing phenotype of *vir-1* mutants. This global phenotype is not restricted to genes that function in defence responses or to those transcripts predicted to have lost m⁶A modification. There is no evidence that immune response genes are more likely to be m⁶A-modified than other mRNAs in the absence of pathogen infection.

## Discussion

We have discovered that disruption of the mRNA m⁶A modification complex triggers autoimmunity in Arabidopsis. Using a combination of Illumina RNA-seq and ONT DRS, we reveal, in unprecedented detail, the temperature-sensitive changes in RNA expression, modification and processing caused by mRNA m⁶A writer complex depletion. At 17°C, most gene expression class changes in *vir-1* mRNA m⁶A writer complex mutants are explained by defence gene activation. By exploiting the established, frequent temperature sensitivity of Arabidopsis immunity, we found that the vast majority of these gene expression changes did not occur when we grew the *vir-1* mRNA m⁶A writer mutant at 27°C. We used orthogonal experimental approaches to examine the biological impact of this defence gene activation. We found temperature-sensitive enhanced levels of premature cell death and resistance to *P. syringae* Pst DC3000 pathogen infection in *vir-1* mutants consistent with the idea that these gene expression changes convert into functional consequences for immunity. Not all gene expression changes, or all developmental phenotypes of *vir-1* mutants, were rescued by elevated temperature, demonstrating that the impacts of m⁶A-dependent changes on gene expression are separable. Together, these different lines of evidence all point to the disruption of the mRNA m⁶A writer complex triggering autoimmunity in Arabidopsis.

The autoimmune phenotypes of Arabidopsis m⁶A writer complex mutants have not previously been described. However, while analysing the impact of mRNA m⁶A on plant pathogen infection, it was recently reported that Arabidopsis mutants defective in METTL3, VIR and WTAP orthologs, and a line ectopically overexpressing the mRNA m⁶A demethylase ALKBH10B, were all more resistant to infection by *P. syringae* DC3000, *P. syringae pv maculicola* and a fungal pathogen *Botrytis cinerea* [85]. Consistent with this, genes commonly upregulated in METTL3 ortholog mutant and ectopic ALKBH10B expression lines are enriched in GO term annotations for defence signalling [85]. Furthermore, early microarray analysis of METTL3 ortholog mutants reported a general enrichment among the overexpressed genes for GO terms related to stress responses [86]. These independent findings are consistent with our analysis of *fip37–4* and *vir-1* and generalise the idea that autoimmunity is a major Arabidopsis mRNA m⁶A writer complex mutant phenotype. We did not detect *PR1* activation in the *te234* triple mutant, which is defective in the function of the most abundant m⁶A reader proteins. However, stress response activation in *te234* has been previously clearly described [87].

The discoveries reported here raise the key question of how the disruption of mRNA m⁶A writer complexes triggers autoimmunity. Our analyses do not explain how defence pathways are autoactivated in Arabidopsis mRNA m⁶A writer complex mutants. Remarkably, given its importance, the mechanism by which modified RNAs ablate TLR signalling in humans has been elusive until recently [88]. We first considered whether m⁶A readers might mediate the autoactivation of defence gene expression. However, we found no evidence that either *cpsf30-yth* mutants or *te234* mutants phenocopied the defence gene expression of *vir-1* mutants. There are 13 m⁶A-reading YTH domain-containing proteins encoded in the Arabidopsis genome. Therefore, approaches that resolve redundancy in their function [82] will be required to further test an association with autoimmunity. The major temperature-sensitive RNA phenotype we detected in *vir-1* mutants was a global change in mRNA poly(A) tail length: we found relatively fewer RNAs with short poly(A) tails (<40 nt) and relatively more with longer poly(A) tails (>100nt) compared to WT Col-0 at 17°C but this phenotype was reversed at 27°C. This

finding could indicate that the trimming of longer poly(A) tails - an essential feature of newly transcribed mRNAs [84] - is defective and/or mRNAs with short poly(A) tails are more susceptible to decay in *vir-1* mutants at 17°C. Poly(A) tails can reduce innate immune responses of human cells to RNA [89]. Furthermore, autoimmunity and pleiotropic developmental defects are phenotypes of Arabidopsis mutants defective in the nuclear poly(A) polymerase, PAP1, that catalyses mRNA poly(A) tail formation [90]. However, the interrelatedness of the phenotypes we detect here is not yet clear. Poly(A) tail length changes were not restricted to transcripts with predicted m⁶A sites, indicating this change is not a direct result of m⁶A loss. Nor were poly(A) tail length changes restricted to transcripts involved in defence functions. Increased poly(A) tail length is a stress response phenotype of different species [91,92] and promotes stress granule formation in humans [92,93]. Therefore, the temperature-sensitive poly(A) tail length changes we detect here may be a previously unappreciated autoimmunity phenotype. An aspect of RNA biology we have not explored is whether changes in condensate association caused by loss of mRNA m⁶A might trigger autoimmunity. mRNA m⁶A modifications can influence separation into biomolecular condensates such as human stress granules [94]. Significantly, the buffering of self RNA by condensates regulates human innate immune responses [95]. Analysing other proteins more closely connected to Arabidopsis poly(A) tail processing and RNA fate [96] could help unravel connections between RNA modification, poly(A) tail length, altered RNA homeostasis and the causes or consequences of autoimmunity.

Different receptors detect RNA as a molecular signature of pathogen infection in humans, and RNA's availability, localisation, and structure (including sequence and modification) are essential criteria for distinguishing self and non-self [97,4]. For example, uridine mononucleotides and di or trinucleotides are bound on different sites of TLR8 in monocyte endosomes in a manner that depends on upstream RNAse 2 and T2 processing of pathogen RNA [98]. RNAs purified from *P. syringae* DC3000 bacteria or transcribed *in vitro* and infiltrated into Arabidopsis leaves activate innate immunity, demonstrating that non-self RNAs can trigger immune responses in Arabidopsis [99]. Aside from the RNAi machinery, which plays crucial roles in viral defence in plants [100], receptors that recognise pathogen RNAs are poorly characterised in plants. However, plant TIR domains have recently been found to hydrolyse RNA [59]. Compared to DNA, RNA is the preferred substrate for TIR domains to synthesise 2',3',-cAMP/cGMP molecules that signal cell death in the hypersensitive response [59]. Crucially, a mutation that disrupts this synthetase activity is sufficient to block cell death signalling [59]. TIR domains are frequently found in Arabidopsis NLR proteins and can also be expressed as TIR-only proteins [101] encoded by TIR-only genes or the widespread prematurely terminated transcripts of NLR genes [62]. We found that mRNAs encoding TIR domain proteins, including TX0, for which RNA hydrolysing activity has been demonstrated [59], were significantly upregulated in *vir-1* mutants. The natural RNA substrates of Arabidopsis TIR domains are unknown. An important question is whether TIR domains sense non-self RNAs or perturbed RNA homeostasis that indicates pathogen activity. The nudix hydrolase family member NUDT7 acts as a phosphodiesterase to modify 2'3'-cAMP/cGMP and thus modulates signalling through EDS1 [59]. Notably, one of the most consistently upregulated genes in *vir-1* is the uncharacterised nudix hydrolase, AtNUDT24.

We stress that since we do not know the mechanism by which defence gene expression is activated in *vir-1* mutants, we have no favoured model. Indeed, the phenomena we report here may be quite indirect: First, it may not be the loss of mRNA m⁶A itself that triggers autoimmunity. ETI functions to detect pathogen activity that disrupts host cell proteins or processes and activates an immune response. In this way, NLRs act as guards, with the effector-targeted host cell proteins or processes being guardees [102,103,104]. It is possible that the mRNA m⁶A writer complex is a guardee and that disruption of the writer complex, rather than the absence of mRNA m⁶A, is detected and triggers autoimmune signalling. Therefore, defence signalling pathways in *vir-1* mutants may directly detect non-modified RNA, a disrupted mRNA m⁶A writer complex, poly(A) tail perturbation, or changed RNA homeostasis resulting from decay or condensate association. Second, the writer complex mutants may disrupt an m⁶A-independent function of writer complex components that triggers autoimmunity. Precedent for this idea comes from the analysis of the *N6*-methyladenosine methyltrasferase, Ime4, subunit of the mRNA m⁶A writer complex of *Saccharomyces cerevisiae*, which has m⁶A-independent roles [105]. Finally, the connection between disrupted m⁶A writer complex and autoimmunity may be even more indirect. Loss of m⁶A

is associated with diverse changes in gene expression and pleiotropic developmental changes. Therefore, if the changes in gene expression, RNA processing, or development found in mRNA m$^6$A writer complex mutants phenocopy features of pathogen infection, they may indirectly trigger immune pathway activation. Consequently, understanding how immune gene expression is activated is crucial to understanding the direct impact of mRNA m$^6$A on these diverse RNA and developmental phenotypes.

No established definition of Arabidopsis autoimmunity exists beyond constitutively activated defence responses in the absence of pathogen infection [106]. Furthermore, no unified signature of gene expression changes that constitutes autoimmunity exists either. Indeed, the recent identification of at least two classes of autoimmunity (microbiota-dependent versus microbiota-independent) in plants [106] reveals we have much to learn about autoimmunity phenomena. Since we identified autoimmunity in *vir-1* mutants grown in sterile conditions, *vir-1* autoimmunity falls into the microbiota-independent class. The comprehensive analysis of gene expression patterns in different autoimmune mutants has the potential to not only define classes of autoimmunity but also to resolve defence gene transcription cascades without the complication of pathogen effector-triggered modifications. Immune responses to localised pathogen attacks involve cell-specific gene expression programmes in infected cells, neighbouring bystander cells and distal tissues [32,27,107]. The molecular details of these cell-specific changes have only recently been explored, but autoimmune mutants have not been examined in this manner. Given that mRNA m$^6$A levels are presumably compromised in writer complex mutants throughout development, it will be interesting to determine in which cells and at what timescales autoactivation of defence gene expression occurs.

Our findings, therefore, have practical implications for studying the impact of mRNA m$^6$A on plant biology. Since most gene expression changes in mRNA m$^6$A writer complex mutants at lower ambient temperatures are caused by autoactivation of defence gene expression, the interpretation of the effects of mRNA m$^6$A will be complicated by indirect changes that vary according to environmental conditions (such as temperature), which may differ between studies. Mutating defence signalling hubs in mRNA m$^6$A writer complex mutant backgrounds might suppress autoimmunity but not necessarily comprehensively block autoimmune signalling. Our global transcriptome analysis only provides snapshots of gene expression changes in Arabidopsis seedlings. However, understanding how mRNA m$^6$A directly influences mRNA processing and fate and, hence, development and autoimmunity will require alternative experimental approaches. Determining immediate gene expression changes following the shutdown of the mRNA m$^6$A writer complex, using, for example, proteolysis-targeting chimaeras (PROTACS) [108], in defined cell types and developmental contexts may help us understand the direct roles of mRNA m$^6$A.

In conclusion, our study establishes a new conceptual framework for analysing the impact of mRNA m$^6$A on plant biology. The molecular basis of the events that trigger mRNA m$^6$A writer complex-dependent autoimmunity is unknown, but uncovering this should lead to fundamental insights into the role of mRNA m$^6$A in plant biology and how defence gene signalling occurs.

## Materials and methods

### Plant material

Wild-type *Arabidopsis thaliana* accession Col-0 and *te234* [109] were obtained from the Nottingham Arabidopsis Stock Centre. The *vir-1* and VIR-complemented (VIR::GFP-VIR) lines [8] were from K. Růžička, Brno, Czechia; *acd6–1* was from J. Greenberg, Chicago, USA; *fip37–4* was from R. Fray, Nottingham, UK; *fls2c* (SAIL_691C4) [71] was from P. Hemsley, Dundee, UK; *cpsf30-yth* (GK477H04) was from Ł. Szewc, Poznań, Poland.

### Plant growth conditions

Seeds were sown on MS10 medium plates, stratified at 4°C for 2 days, germinated in a controlled environment at 22°C under 16 hr light/8 hr dark conditions and harvested for RNA purification 14 days after transfer to 22°C. For temperature assays, plant growth chambers were set to either 17°C or 27 °C, with all other conditions the same as above. Seedlings

were harvested 14 days after germination during the first two hours of the light period following an 8-hour dark phase. Four-week-old plants were used for phenotyping the adult plants at 17°C or 27 °C.

### Trypan blue staining

Trypan blue staining was performed on leaves from 4-week-old plants of WT Col-0 and *vir-1* grown at 17°C and 27°C. Leaves were stained in a solution of Tris-EDTA equilibrated phenol (pH 8) (25%), glycerol (25% v/v), lactic acid (25% v/v) with trypan blue (10mg/ml). Leaves were treated with staining solution for 10 minutes at 95°C then incubated overnight at room temperature. Leaves were destained in chloral hydrate solution twice, for 4 h and overnight. Stained leaves were imaged under a Zeiss histology microscope at 10x magnification. Images were imported into ImageJ [72], and the total stained area was measured in pixels, with the stained area expressed as a percentage of the total leaf area. Data collected from 10 leaves per condition was plotted, and a two-way ANOVA test with post hoc Turkey's HSD tests was used to assess the effects of genotype and temperature and their interaction on the percentage of leaf stained.

### Pathogenesis assays

Arabidopsis Col-0, *vir-1* and *fls2c* seedlings were treated with *Pseudomonas syringae* pv tomato (Pto) DC3000 using flood inoculation. Bacteria were cultured on MG agar supplemented with rifampicin at 28°C for 24–48 h. Four-week-old seedlings flood-inoculated with a bacterial suspension of *P. syringae* DC3000 ($5x10^6$ Colony-Forming Units (CFU)/ml) containing 0.025% Silwet L-77. Sterilised seedlings were grown on half-strength MS agar for 2–3 weeks at 17°C, 21°C and 27°C before inoculation. The bacterial suspension was applied to the Arabidopsis seedlings, and plates were incubated at room temperature for 2–3 minutes. Excess liquid was drained, and seedlings were maintained at growing temperatures for three days. Bacterial growth was calculated using serial dilution of material from three seedlings per plate and recorded as CFU/mg. The significance of changes in bacterial growth in the differing conditions was tested using ANOVA. The experiment was repeated to confirm results.

### RNA isolation

Total RNA was isolated using the RNeasy Plant Mini kit (Qiagen) and treated with RNAse-free DNase (Promega-M6101). RNA concentration and integrity were measured using a NanoDrop one spectrophotometer and Agilent 4150 Tapestation.

### Gene expression analysis by RT-qPCR

Total RNA was extracted from 14-day-old seedlings. Total RNAs were treated with RNAse-free DNase (Promega-M6101). First-strand cDNAs were synthesised using SuperScript™ III Reverse Transcriptase (Thermo Fisher Scientific-12574026). qPCR was carried out on a LightCycler® 96 Instrument using Brilliant III Ultra-Fast SYBR Green qRT-PCR Master Mix (Agilent-600886). Three biological replicates (independently harvested samples) with three technical replicates for each were analysed. Relative expression levels were determined using the $2^{-\Delta\Delta CT}$ method. Arabidopsis *UBQ10* (AT4G05320) was used as internal control. qPCR primer pairs are listed in S16 File.

### Preparation of libraries for Illumina RNA-sequencing

Illumina RNA-seq libraries were prepared by Genewiz (Azenta LifeScience) using NEB Next Ultra Directional Library Prep Kit according to the manufacturer's instructions. Paired-end sequencing with a read length of 150 bp was carried out on the Illumina NovaSeq X following the manufacturer's instructions. Raw sequence data was converted to fastq and de-multiplexed using Illumina bcl2fastq version 2.20.

## Processing of Illumina RNA-seq data and differential gene expression analysis

Quality assessment of RNA-seq reads was performed using FastQC [110]. For digital expression, the Salmon index was built using Arabidopsis Araport11 transcript annotations [111]. Transcript and gene-level counts were estimated using Salmon (with gtf option and Araport 11 annotation) (version 1.9.0) [112]. Differential expression analysis was performed in edgeR (version 4.2.2) using a quasi-likelihood generalised linear model (glmQLFit). Annotation of genes of interest, categorising them as defence, flowering or other and returning GO annotation with further annotation was performed using custom scripts: https://github.com/bartongroup/PT_Arabidopsis_names_to_annot. To visualise dimensional reduction in the context of RNA-seq quality control, PCA plots were created using ggplot2 (version 3.5.1). Correlation and heatmap plots were generated with the ptr script from the Trinity RNAseq package (version 2.15.2) [113]. GO enrichment heatmap were made using msbio (metascape)(version 3.5.20240901) [114].

Functional enrichment analysis was performed using a combination of Goseq (version 1.42.0) [115] and g:Profiler (version e111_58_p18_f463989d) with the g:SCS multiple testing correction method and a significance threshold < 0.05 [48]. Domain enrichment analysis was performed in DAVID [50] using a FDR significance threshold of < 0.05.

Tests for motif enrichment in promoter regions were carried out using Analysis of Motif Enrichment (AME) (version 5.5.8) in MEME (version 5.5.7). Promoters were defined as the 1.5kb region upstream of the transcript start site. A set of 5000 promoter sequences from randomly selected genes were used as the background.

To determine whether pathogen contamination was present in the *vir-1* RNA-seq samples, reads were mapped to the TAIR10 genome, and from the resulting bam file, unmapped reads were returned using STAR (version 2.7.11b) [116]. BBnorm (October 19, 2017) was then used to normalise the "unmapped" reads with the following setting "target=75 min=3" [117]. The normalised reads were assembled using Trinity [113] (version 2.15.2) with –trimmomatic –no_normalise. The transcriptome assembly was processed using cd-hit-est (version 4.8.1) (-c 0.90 -n 8 -T 24 -M 0) [118] to reduce redundancy at 90%. The resulting final transcriptome assembly was then searched against Genbank NR with Diamond-BLASTP using Diamond (version v2.0.5.143) [119]. The diamond BLASTP output was post-taxonomically annotated using https://github.com/peterthorpe5/public_scripts/tree/master/Diamond_BLAST_add_taxonomic_info. The final taxonomically assigned BLAST output was then interrogated for the presence of plant pathogens, as defined here (https://phytopathdb.org/pathogens_eg/). Digital expression per condition to this assembly was estimated using Salmon [120], and differential expression analysis was performed as described above.

## ONT DRS library preparation

Total RNA was isolated, as detailed above. Poly(A)+ RNA was purified from approximately 100µg of total RNA using the Dynabeads mRNA purification kit (Thermo Fisher Scientific) following the manufacturer's instructions. The quality and quantity of mRNA were assessed using the Nanodrop one spectrophotometer (Thermo Fisher Scientific) and Tape station 4150 (Agilent Technologies). ONT DRS libraries were prepared from 100ng poly(A)+ RNA for the WT Col-0 - *vir-1* comparison at 17°C and 27°C. All other ONT DRS libraries were prepared from 500ng poly(A)+ RNA. Libraries were made using the Direct RNA sequencing kit (SQK-RNA002; Oxford Nanopore Technologies) according to the manufacturer's instructions. The poly(T) adapter was ligated to the mRNA using T4 DNA ligase (New England Biolabs) in the Quick Ligase reaction buffer (New England Biolabs) for 15min at room temperature. First-strand cDNA was synthesised by SuperScript III Reverse Transcriptase (Thermo Fisher Scientific) using the oligo(dT) adapter. The RNA–cDNA hybrid was then purified using Agencourt RNAClean XP magnetic beads (Beckman Coulter). The sequencing adapter was ligated to the mRNA using T4 DNA ligase (New England Biolabs) in the Quick Ligase reaction buffer (New England Biolabs) for 15min at room temperature followed by a second purification step using Agencourt beads (as described above). Libraries were loaded onto R9 version SpotON Flow Cells (Oxford Nanopore Technologies) and sequenced using a GridION device at the Tayside Centre for Genomic Analysis, School of Medicine, University of Dundee, for a 48-hour runtime. Four biological replicates were performed for each genotype.

## ONT DRS mapping

Reads were basecalled using Dorado version 0.5.3 (Oxford Nanopore Technologies) using the rna002_70bps_hac@v3 high accuracy model. Reads were aligned to the Araport11 transcriptome [111] and the TAIR10 Arabidopsis genome [121] using minimap2 version 2.17 [122] cond for spliced alignment. The following parameters were used for both alignments: --end-seed-pen = 15 for end seed penalties, -A1, -B1, -O2,32, -E1,0 and -C9 to tune alignment scoring. For genomic alignment, splice junction information was incorporated using the --junc-bed parameter, which referenced the annotated introns BED file. A junction bonus of 10 (--junc-bonus = 10) was applied to prioritise alignments that utilised known splice junctions, increasing alignment accuracy for spliced reads. The spliced alignment was optimised with parameters -k14, -uf, -w5, --splice, and -g2000, along with a maximum intron size of 200,000 (-G200000), --splice-flank = yes for spliced alignment flanking detection, and -z200 for seeding thresholds. Alignments were converted to BAM files and indexed using samtools version 1.18.

## Prediction of m⁶A in ONT DRS data using m6Anet

Event information was extracted from raw signal data and transcriptome alignments using the f5c implementation of eventalign with event scaling [123]. Aligned event files were processed using m6anet [75]. Data preparation and inference was performed using the pretrained Arabidopsis model with the default read probability threshold of <0.0033. Predicted sites of modification were filtered using the recommended probability-modified threshold of >0.9 [75]. The distribution of predicted modification ratios for all sites passing this threshold was plotted for each condition.

## Analysis of poly(A) site usage in ONT DRS data

Differential 3' analysis was performed on bam files using the d3pendr tool as described previously [62]. The statistical significance of the 3' shift was assessed by permuting read alignments between the control and treatment distributions to determine the maximum distance achieved by random sampling.

## Estimation of poly(A) tail length in ONT DRS data

The length of poly(A) tails per read was estimated using "--no-trim --estimate-poly-a" with the following model: rna002_70bps_hac@v3 in Dorado (version 0.5.3). Reads were mapped to the Araport 11 transcriptome using minimap2 (see above), and the resulting bam file was used to generate a read-to transcript table for further interrogation. Differences in mean poly(A) tail length per gene between conditions were calculated as previously described [12]. In brief, poly(A) tail lengths were aggregated by gene ID, and where genes were present with at least ten reads in both conditions, the distributions of poly(A) tail lengths were compared using the Wasserstein distance. Significance was assessed using a permutation test with 999 bootstraps. Genes were classed as m⁶A-modified if they had at least one site above the probability-modified threshold of >0.9 in at least one Col-0 sample.

## Gene tracks

Gene track figures were generated using Matplotlib (version 3.9.2) from normalised bigwig files of Illumina RNA-Seq coverage and pooled bam files of reads per condition. For tracks with >100 ONT DRS reads per condition, a random subsample of 100 reads per track was plotted.

## m⁶A liquid chromatography with tandem mass spectrometry

m⁶A analysis using tandem liquid chromatography-mass spectroscopy (LC/MS-MS) was performed as previously described [12,62]. LC/MS-MS was carried out by the FingerPrints Proteomics facility at the University of Dundee. A two-way ANOVA test was used to assess the effects of genotype and temperature and their interaction on the ratio of m⁶A to A.

## Supporting information

**S1 Fig. Immune gene expression is activated in m⁶A writer complex mutants.** A) Normalised log2 counts per million of *PR1* (AT2G14610) in Col-0, Col-0, *vir-1* and VIRc in ONT DRS reads (n = 4 samples per genotype). B) Upregulation of *PR1* (AT2G14610) in *vir-1* at 20°C, shown by a gene track of Illumina RNA-seq and downsampled ONT DRS reads.
(TIF)

**S2 Fig. A) Normalised logged counts per million of PR1 (AT2G14610) in Col-0 and vir-1 17°C and 27ºC (n = 3–4 per condition).** B) RT-qPCR showing the upregulation of *PR1* in *vir-1* at 17°C (n = 3 per condition). C) Volcano plot showing the log2 fold change and adjusted p-value of differential gene expression in *vir-1* at 17°C contrasted to the average expression in *vir-1* at 27°C and Col-0 at 17°C and 27°C. Genes with log2FC > 2 and p < 0.001 are coloured in red, genes which only pass the p-value threshold are coloured in black, and genes which only pass the log2FC threshold are coloured in blue. Non-significant changes (NS) are coloured in grey. Source data available in S5 File. D) Overlap in enriched GO terms between genes upregulated at 17°C contrasted to the average expression in *vir-1* at 27°C and Col-0 at 17°C and 27°C, and genes which were significantly upregulated in *vir-1* at 22ºC contrasted to Col-0 at 22ºC.
(TIF)

**S3 Fig. A) Upregulation of *SARD1* (AT1G73805) in *vir-1* at 17°C, shown by a boxplot of normalised expression (log 2 counts per million) in ONT DRS data (n = 3–4 samples per condition).** B) Upregulation of *FLS2* (AT5G46330) in *vir-1* at 17°C, shown by a boxplot of normalised expression (log 2 counts per million) in ONT DRS data (n = 3–4 samples per condition). C) Upregulation of *TX10* (AT1G57630) in *vir-1* at 17°C, shown by a boxplot of normalised expression (log 2 counts per million) in ONT DRS data (n = 3–4 samples per condition). D) Gene track of ONT DRS data showing the upregulation of a novel TIR domain-containing gene (annotated as Novel gene) downstream of *RPS6* (AT5G46470) in *vir-1* at 17°C (n = 3–4 samples per condition). E) Upregulation of *ACD6* (AT4G14400) in *vir-1* at 17°C, shown by a boxplot of normalised expression (log 2 counts per million) in ONT DRS data (n = 4 samples per condition). F-M) Boxplots showing the normalised log 2 counts per million (as produced by edgeR) for the flowering genes; *FT* (AT1G65480), *FUL* (AT5G60910), *SOC1* (AT2G45660), *SEP3* (AT1G24260), *SPL4* (AT1G53160), *SPL5* (AT3G15270), *AGL19* (AT4G22950) and *AGL24* (AT4G24540), in Illumina RNA-seq of *vir-1* and Col-0 at 17ºC and 27ºC (n = 4 samples per condition).
(TIF)

**S4 Fig. Four-week-old Col-0 WT, *vir-1* and *fls2c* seedlings flood-inoculated with a bacterial suspension of *Pst* DC3000 (5x10⁶ CFU/ml) and 0.025% v/v Silwet L-77.** Bacterial populations were quantified at 3 days post-inoculation (dpi) (n = 3 per condition). One way ANOVA tests on each genotype revealed a significant effect of temperature in the *vir-1* genotype (F = 23.02, p = 0.00197) which was not present in Col-0 WT or *fls2*. Source data available in S9 File. This experimental analysis represents an independent replication of the experiment presented in Fig 4A.
(TIF)

**S5 Fig. A) Gene track showing the expression of *VIRILIZER* in Illumina RNA-seq *VIR* expression in *vir-1* mutants at 17°C and 27°C.** B) Magnified portion of the *VIRILIZER* gene track showing the combined coverage and alignment of Illumina RNA-seq around the EMS point mutation in *vir-1*. The *vir-1* mutation affects the 5' splice site of intron 5 (G + 1 to A), which leads to the activation of cryptic 5' splice sites upstream in exon 5 detected only in *vir-1* (denoted by an arrow). No suppression of this cryptic splicing is found at 27 °C. Aligned reads were subsampled to 200 reads per condition. C) Density distribution of the ratio of modification predicted by Yanocomp, for modifications with an FDR < 0.05. Predicted modification ratios for *vir-1* and Col-0 at 17°C were obtained by comparisons of *vir-1* at 17°C and Col-0 at 17°C. Predicted modification ratios for *vir-1* and Col-0 at 27°C were obtained by comparing *vir-1* at 27°C and Col-0 at 27°C.
(TIF)

**S6 Fig.  A) Phenotype of Col-0, _vir-1, fip37–4, cpsf30-yth, te234_ and _acd6–1_ grown at 17ºC and 27ºC.** B) Gene ontology biological process terms enriched in the 58 genes consistently significantly upregulated (log2FC+/- 2.0 FDR<0.001) in _vir-1_ across 17°C, 20°C and 27°C compared to Col-0 and VIRc. Source data available in S13 File. C) Upregulation of _FMO1_ (AT1G19250) in _vir-1_ at both 17°C and 27°C in Illumina RNA-seq and ONT DRS data, shown by gene tracks and boxplots of normalised expression (log 2 counts per million).
(TIF)

**S7 Fig.  A) Density distribution of poly(A) tail lengths of _Saccharomyces cerevisiae ENOLASE II_ spike-in sequences in Col-0 and _vir-1_ at 17°C and 27°C.** _ENOLASE II_ transcripts with an estimated poly(A) tail length of 30 nt are included as the RNA calibration standard during ONT DRS library preparation. B) Density distribution of estimated poly(A) tail lengths for reads aligning to _GAPC2_ (AT1G13440) in Col-0 and _vir-1_ at 17°C and 27°C C) Density distribution of estimated poly(A) tail lengths for reads aligning to _UBQ10_ (AT4G05320) in Col-0 and _vir-1_ at 17°C and 27°C D) Density distribution of estimated poly(A) tail lengths for reads aligning to _CAB1_ (AT1G29930) in Col-0 and _vir-1_ at 17°C and 27°C E) Density distribution of estimated poly(A) tail lengths for reads aligning to genes with significantly higher (logFC>2.0, FDR<0.001) expression in vir-1 at 17ºC compared to other conditions F) Density distribution of estimated poly(A) tails in Col-0 and _vir-1_ at 17°C and 27°C, divided into those belonging to genes with a predicted m$^6$A modification in Col-0 and those with no predicted modification. The distribution of estimated poly(A) tails is plotted individually for each replicate.
(TIF)

**S1 File.  Sequencing statistics for ONT DRS and Illumina RNA sequencing of _vir-1_ and Col-0 at 17ºC and 27ºC.**
(XLSX)

**S2 File.  Differential Gene Expression Results from Illumina RNA sequencing at 22ºC.**
(XLSX)

**S3 File.  Differential Gene Expression Results from ONT direct RNA sequencing of _fip37–4_ at 22ºC.**
(XLSX)

**S4 File.  Differential Gene Expression Results from Illumina RNA sequencing of _vir-1_ and Col-0 at 17ºC.**
(XLSX)

**S5 File.  Differential Gene Expression Results from Illumina RNA sequencing of _vir-1_ at 17ºC.**
(XLSX)

**S6 File.  GO terms and protein domains enriched in genes upregulated in _vir-1_ at 17ºC compared to other conditions.**
(XLSX)

**S7 File.  Significantly enriched motifs found using MEME AME.**
(XLSX)

**S8 File.  Blast hits for pathogen reads detected in _vir-1_ and Col-0 samples.**
(XLSX)

**S9 File.  Defence-related and flowering genes upregulated in _vir-1_.**
(XLSX)

**S10 File.  Flood Innoculation of Arabidopsis with _Pseudomonas syringae_ pv tomato (Pto) DC3000.**
(XLSX)

**S11 File. LC/MS-MS results for *vir-1* and Col-0 samples.**
(XLSX)

**S12 File. Quantified Trypan blue staining data.**
(XLSX)

**S13 File. Overlap in gene upregulation in *vir-1* compared to Col-0 at 17ºC, 22ºC and 27ºC.**
(XLSX)

**S14 File. Gene Ontology terms enriched in consistently upregulated genes.**
(XLSX)

**S15 File. Genes with significant changes in poly(A) tail length between conditions.**
(XLSX)

**S16 File. Primers used in this study.**
(XLSX)

## Acknowledgments

We thank Prof. Steven Spoel (University of Edinburgh) for *Pseudomonas syringae* pv. Tomato (*Pto*) DC3000. We thank Dr. Martin Balcerowicz (University of Dundee) for providing access to temperature-controlled environment cabinets and Dr. Rachel Taylor (University of Leeds) for temperature-controlled plant growth. We are grateful to Katie Dempsey, whose experiments led us to investigate the temperature sensitivity of *vir-1* mutants. We thank Dr. Martin Balcerowicz, Prof. Brendan Davies and Dr. Davide Bulgarelli for helpful comments on the manuscript. We thank the University of Dundee HPC and Research Computing at the James Hutton Institute for providing computational resources and technical support through the BBSRC-funded "UK's Crop Diversity Bioinformatics HPC" (BB/S019669/1 and BB/X019683/1). The FingerPrints Proteomics facility at the University of Dundee is supported by a Wellcome Trust Technology Platform Award (097945/B/11/Z).

## Author contributions

**Conceptualization:** Carey L Metheringham, Anjil K Srivastava, Peter Thorpe, Matthew Parker, Gordon Simpson.

**Data curation:** Carey L Metheringham, Peter Thorpe.

**Formal analysis:** Carey L Metheringham, Anjil K Srivastava, Peter Thorpe, Ankita Maji, Matthew Parker, Gordon Simpson.

**Funding acquisition:** Geoffrey Barton, Gordon Simpson.

**Investigation:** Carey L Metheringham, Anjil K Srivastava, Peter Thorpe, Ankita Maji, Matthew Parker.

**Methodology:** Carey L Metheringham, Anjil K Srivastava, Peter Thorpe.

**Project administration:** Geoffrey Barton, Gordon Simpson.

**Resources:** Geoffrey Barton, Gordon Simpson.

**Software:** Carey L Metheringham, Peter Thorpe.

**Supervision:** Geoffrey Barton, Gordon Simpson.

**Validation:** Anjil K Srivastava, Peter Thorpe, Matthew Parker.

**Visualization:** Carey L Metheringham, Peter Thorpe, Ankita Maji, Matthew Parker.

**Writing – original draft:** Carey L Metheringham, Anjil K Srivastava, Peter Thorpe, Ankita Maji, Gordon Simpson.

**Writing – review & editing:** Carey L Metheringham, Anjil K Srivastava, Peter Thorpe, Ankita Maji, Matthew Parker, Geoffrey Barton, Gordon Simpson.

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
