## [Decision Letter · Decision Letter 0]

14 Oct 2025

Dear Dr Simpson,

Your revised manuscript has been reviewed by two of the original reviewers, whose comments are included below. Based on the reviewers' comments, we are pleased to inform you that your manuscript entitled "Disruption of the mRNA m6A writer complex triggers autoimmunity in Arabidopsis" has been editorially accepted for publication in PLOS Genetics. Congratulations!

Yours sincerely,

Chunxiao Song

Academic Editor

PLOS Genetics

Paula Cohen

Section Editor

PLOS Genetics

Aimée Dudley

Editor-in-Chief

PLOS Genetics

Anne Goriely

Editor-in-Chief

PLOS Genetics

BlueSky: @plos.bsky.social

Comments from the reviewers (if applicable):

Reviewer's Responses to Questions

**Comments to the Authors:**

Reviewer #1: The authors have thoroughly addressed my comments and revised the manuscript accordingly. I appreciate the in-depth presentation of the poly-A tail length phenotype. The authors have clarified in the text that there is no evidence to connect the polyA tail length phenotype to m6A modification in the transcripts themselves. I agree with the authors that it's a striking phenotype worth presenting in the literature. The discussion is quite good.

Reviewer #2: In their revised version, the authors have incorporated several improvements that satisfy the (relatively minor) concerns I raised on the originally submitted version.

I am happy for the manuscript to be accepted for publication in its current state.

**Data Deposition**

http://datadryad.org/submit?journalID=pgenetics&manu=PGENETICS-D-25-00947

**Press Queries**

---

## [Editor Report · Acceptance letter]

PGENETICS-D-25-00947

Disruption of the mRNA m6A writer complex triggers autoimmunity in Arabidopsis

Dear Dr Simpson,

We are pleased to inform you that your manuscript entitled "Disruption of the mRNA m6A writer complex triggers autoimmunity in Arabidopsis" has been formally accepted for publication in PLOS Genetics! Your manuscript is now with our production department and you will be notified of the publication date in due course.

With kind regards,

Anita Estes

PLOS Genetics

On behalf of:
